# Risk Stratification for Immediate Postoperative Hearing Loss by Preoperative BAER (Brainstem Auditory Evoked Response) and Audiometry in NF2-Associated Vestibular Schwannomas

**DOI:** 10.3390/cancers13061384

**Published:** 2021-03-18

**Authors:** Isabel Gugel, Florian Grimm, Philip Hartjen, Maria Breun, Julian Zipfel, Marina Liebsch, Hubert Löwenheim, Ulrike Ernemann, Lan Kluwe, Victor-Felix Mautner, Marcos Tatagiba, Martin Ulrich Schuhmann

**Affiliations:** 1Department of Neurosurgery and Neurotechnology, University Hospital Tübingen, BW 72076 Tübingen, Germany; florian.grimm@med.uni-tuebingen.de (F.G.); julian.zipfel@med.uni-tuebingen.de (J.Z.); marina.liebsch@med.uni-tuebingen.de (M.L.); marcos.tatagiba@med.uni-tuebingen.de (M.T.); martin.schuhmann@med.uni-tuebingen.de (M.U.S.); 2Centre of Neurofibromatosis and Rare Diseases, University Hospital Tübingen, BW 72076 Tübingen, Germany; v.mautner@uke.de; 3Department of Neurology, University Medical Center Hamburg-Eppendorf, HH 20251 Hamburg, Germany; p.hartjen@uke.de (P.H.); kluwe@uke.de (L.K.); 4Department of Maxillofacial Surgery, University Medical Center Hamburg-Eppendorf, HH 20251 Hamburg, Germany; 5Department of Neurosurgery, University Hospital Würzburg, BY 97080 Würzburg, Germany; breun_m@ukw.de; 6Division of Pediatric Neurosurgery, University Hospital Tübingen, BW 72076 Tübingen, Germany; 7Department of Otorhinolaryngology, Head and Neck Surgery, Tübingen Hearing Research Center, University Hospital Tübingen, BW, Elfriede-Aulhorn-Straße 5, BW 72076 Tübingen, Germany; hubert.loewenheim@med.uni-tuebingen.de; 8Department of Neuroradiology, University Hospital Tübingen, BW 72076 Tübingen, Germany; ulrike.ernemann@med.uni-tuebingen.de

**Keywords:** hearing preservation, neurofibromatosis type 2, vestibular schwannoma, audiometry, brainstem auditory evoked potentials

## Abstract

**Simple Summary:**

Hearing preservation is one of the major goals in the surgical treatment of Neurofibromatosis Type 2 (NF2) associated vestibular schwannomas (VS) and hearing diagnostics are important monitoring parameters and intraoperative tools to pursue this goal. Our monocentric retrospective analysis aimed to predict postoperative hearing deterioration or loss based on preoperative audiometry and neurophysiological (brainstem auditory evoked potentials, BAEP) findings. In this respect and according to our data evaluation in 100 NF2-associated VS of 72 patients both parameters seem to be prognostic markers, particularly BAEP findings. Large discrepancies between both audiometry and BAEPs which were identified in twenty-one cases of our cohort appear to be at high risk of significant postoperative hearing worsening.

**Abstract:**

Both brainstem auditory evoked potentials (BAEP) and audiometry play a crucial role in neuro-oncological treatment decisions in Neurofibromatosis Type 2 associated (NF2) vestibular schwannoma (VS) as hearing preservation is the major goal. In this study, we investigated the risk of immediate postoperative hearing deterioration (>15 dB and/or 15% loss in pure-tone average [PTA]/ speech discrimination score [SDS] in a cohort of 100 operated VS (ears) in 72 NF2 patients by retrospective analysis of pre- and postoperative hearing data (PTA, SDS, American Association of Otolaryngology–Head and Neck Surgery [AAO-HNS], and brainstem auditory evoked potential [BAEP] class) taking into account relevant influencing factors, particularly preoperative audiometry and BAEP status and the extent of resection. Immediately after surgery, the hearing was preserved in 73% of ears and approximately ~60% of ears kept their hearing classes. Preoperative BAEP (*p* = 0.015) and resection amount (*p* = 0.048) significantly influenced postoperative hearing outcome. The prediction model for postoperative hearing deterioration/loss between preoperative BAEP and AAO-HNS class showed increased risk by increasing BAEP class. Twenty-one tumors/ears were identified with large BAEP and AAO-HNS class discrepancies (≥2 points) and were associated with a high (48–100%) risk of deafness after surgery in ears with preoperative available hearing. Overall, the results were heterogeneous but the better both BAEP and audiometry class before surgery, the higher the chance of hearing maintenance afterwards. Large resection amounts (e.g., 100% risk in near-total resections) exhibit a significant (*p* < 0.05) higher risk compared to smaller amounts (e.g., 10/20% in laser-coagulated/partially resected tumors). Our results emphasized the indispensable role of both hearing monitoring in form of audiometry and neurophysiology (BAEP) in the pre-and perioperative monitoring of NF2-associated VS. Both BAEP and audiometry are good prognostic markers for the postoperative hearing outcome. The extent of resection should be strictly guided by and adjusted to the intraoperative neurophysiological monitoring.

## 1. Introduction

Neurofibromatosis type 2 (NF2) is an autosomal-dominant tumor predisposition syndrome caused by the inactivation of the *NF2* gene located on chromosome 22q [1,2]. The typical hallmark of the disease is the presence of bilateral vestibular schwannomas (VSs) [3,4,5]. Patients also present with other comorbidities (e.g., intracranial, or spinal meningiomas or schwannomas, juvenile cataracts, or brainstem stroke) [3,4,5,6,7] and genotype-phenotype correlations are well described [8,9]. Nevertheless, the course of the disease always remains individual and hardly predictable particularly for bilateral VS. Due to their presence the risk of bilateral deafness is ubiquitous and is long in the focus of attention in the management of the disease, particularly during childhood and adolescence.

Treatment options in NF2-associated VS are currently limited to surgery [10,11], chemotherapy, especially with bevacizumab [12,13], and radiosurgery [14], and can be combined during the course of disease [15,16]. The effects of adjuvant bevacizumab treatment in young NF2 patients regarding growth control are heterogeneous and hearing control does not achieve significant response results [16]. Surgical therapy for NF2-associated VS should always aim at functional preservation, particularly of hearing. This often implies smaller resection amounts, since total resection amounts carry a high risk of surgery-induced deafness. Thereby neurophysiological monitoring in form of brainstem auditory evoked potentials (BAEPs) plays a very important role both in the monitoring of hearing during the course of the disease, for treatment decisions, and of course intraoperatively [17,18]. It is the most sensitive criterion for monitoring of hearing, and a deterioration in pure-tone and speech audiometry and therefore functional hearing is often preceded by an electrophysiological deterioration. In the majority of cases, these parameters correlate with each other, as we were able to show in a previous study [10]. Nevertheless, this positive correlation cannot always be observed and the combined predictive value of BAEP and audiometry for outcome after surgical intervention is not fully understood.

In the following study, we investigated the combined role of preoperative BAEP and audiometry on postoperative hearing function and evaluated the risk of postoperative hearing decline or loss in ears with large discrepancies between preoperative BAEP and audiometry.

## 2. Materials and Methods

### 2.1. Patients and Clinics

The diagnosis of NF2 was confirmed in all patients by clinical evaluation using the Baser diagnostic criteria for NF2 [19].

A total of 72 NF2 patients and 100 tumors with complete hearing diagnostic before and after surgery were included in this retrospective analysis and who were followed-up between 2004 and 2019, at the Department of Neurosurgery and Centre of Neurofibromatosis in Tübingen. The Ethics Board of the Medical Faculty and the University Hospital of Tübingen approved this retrospective analysis (No 018/2019BO2, approval date 17 January 2019). Surgery was performed at our institution via the retrosigmoid approach, by decompression of the internal auditory canal (IAC) with tumor various resection amounts, under continuous neurophysiological monitoring and only by the two senior authors, both very experienced in VS surgery (M.T., M.U.S.).

The indications for surgery were (1) hearing deterioration in brainstem auditory evoked potentials (BAEP) and/or pure-tone and/or speech audiometry regardless of tumor size (except very small and purely intrameatal tumors not reachable for bony decompression of the IAC) often associated with tumor growth, and (2) large tumors (T4, Hannover Classification System [20]) on both sides with brainstem compression. Solid tumor growth in small to middle-sized (T1–T3) tumors with normal/stable hearing were no indication for surgery.

Data in the observation course, under bevacizumab treatment, after radiation or with hearing aids were excluded. Mutation analysis was performed during routine diagnostic in 48 patients. The remaining 24 patients or their legal guardians refused genetic analysis.

Data of 37 (of 100) tumors were part of a prior study [10] and were included in this cohort.

### 2.2. Volumetry, Growth Rate, and Hearing

Tumor volumetry, growth rate, and resection amount were measured, classified, and calculated as previously described [10,11].

For the volumetric analysis, postcontrast thin-sliced (≤3 mm) T1-weighted magnetic resonance (MR) images were uploaded into the iPlan Net software (Brainlab, Feldkirchen, Germany) and volumes were measured by one experienced rater (I.G.) using the manual segmentation tool.

The growth rate was determined using MRI data sets from at least 2-time points before and after surgery. Postoperatively, the growth rate determination was restarted from the first MRI session performed 3 months after surgery.

The extent of resection was calculated as a percentage amount from pre-and postoperative volume values. For this calculation, there was a lack either of preoperative or postoperative volume values in 5 (out of 100) tumors. In addition, a categorical classification was also made analogous to our previous study [11].

Hearing was assessed by regular determination of 4-frequency pure-tone average (PTA), speech discrimination score (SDS), and BAEP, within 4 weeks before surgery and directly or up to 3 months after surgery. Hearing data were classified using the Gardner and Robertson Scale (G–R) [21], the American Association of Otolaryngology–Head and Neck Surgery (AAO–HNS) [22], and BAEP Classification System [18].

Immediate postoperative functional hearing deterioration was defined as a decrease of >15 dB in PTA and/or >15% in SDS compared to preoperative values and as a downgrading of G-R, AAO-HNS, and BAEP class.

Neurophysiological monitoring in form of BAEPs was performed with the Nicolet^TM^ Viking Quest system (Natus Medical Incorporated, San Carlos, CA, USA) in all patients according to a standardized protocol. Monitoring was performed with condensation, rarefaction, and alternate polarity as continuous click stimulation of 115 dB pSPL intensity, generated by an audio stimulator driven by pulses of 100µs duration. The stimuli were presented through TIP-100 tubal earphones inserted into the external auditory canal with a stimulus repetition rate of 10.3 Hz. The contralateral ear was masked by 80 dB white noise. A double measurement was derived for both sides with 1000 to 1500 sweeps and filter settings of 150–15,000 Hz. The latencies of waves I, III, and V as well as the interpeak latencies I–III, III–V, and I–V as well as the amplitude quotient of wave V/I were determined. During the examination, the patient is in a lying position, has its eyes closed, and is ideally maximally relaxed. The intraoperative multimodal neurophysiological monitoring additionally included facial electromyography and long tract sensory evoked potentials.

BAEPs were categorized using the classification system by the Hannover group [18] which is illustrated in Table 1.

### 2.3. Data Evaluation

Statistical evaluation was performed with Statistical Package for Social Sciences (SPSS) (IBM Corp. SPSS Statistics for Windows, Version 22.0. Armonk, NY, USA) and Matlab (MATLAB, (2019), version 9.5.0 (R2019b), Natick, Massachusetts: The MathWorks Inc., USA).

Binomial logistic regression was performed to ascertain the effects of preoperative tumor volume, growth rate, PTA, SDS, AAO–HNS, G–R, and BAEP class as well as resection amounts and age at time of diagnosis on the likelihood that participants suffer from postoperative hearing loss.

The linearity of the continuous variables with respect to the logit of the dependent variable was assessed via the Box–Tidwell procedure. Based on this assessment, all continuous independent variables were found to be linearly related to the logit of the dependent variable. There were five standardized residuals (tumor numbers 1, 27, 47, 49, and 67) with values of 3.384, 1.735, 3.492, −2.179, and −1.094 standard deviations, which were kept in the analysis.

Risk stratification of postoperative hearing deterioration was estimated according to the preoperative AAO–HNS und BAEP scale.

A Chi-square test of homogeneity was performed to investigate the relation of resection amount categories (1 to 7, description in Table 2) and the risk of direct postoperative hearing deterioration or loss. On significant differences, a post hoc analysis was applied involving pairwise comparisons using the z-test and two proportions with a Bonferroni correction.

## 3. Results

### 3.1. Patients, Tumors, and Clinics

Detailed demographic and clinical data are summarized in Table 2 and Table 3 and the clinical course of seven interesting exemplary cases with large discrepancies (≥2 points) between BAEP and audiometry is shown in the Appendix A. Patient/tumor-specific information of the overall cohort is attached in the Appendix A.

Nearly two-thirds of tumors (61%) were partially resected. 8/48 patients were ascertained to be mosaic and in 4/48 patients no mutation could be detected neither in blood nor in tumor DNA. 25 patients were treated with bevacizumab in the following course due to further hearing deterioration and/or tumor growth.

### 3.2. Immediate Postoperative Hearing Function

A total of 73% (*n* = 73/100) of ears maintained and 10% (*n* = 10/100) were unable to maintain their hearing (PTA and SDS) according to hearing preservation criteria. Deafness (=PTA 130 dB and 0% SDS) occurred in 17 ears (17%) after surgery.

Preservation of the preoperative hearing grade could be achieved in 61 (61%), 59 (59%), and 65 (65%) of hearing ears, with regards to the Gardner and Robertson Scale (G–R), American Association of Otolaryngology–Head and Neck Surgery (AAO–HNS) and the BAEP classification, respectively. An improvement in G–R Scale, AAO–HNS, and BAEP class could be observed in two, one and four ear/ears each.

Overall functional hearing (AAO–HNS class I and II) could be preserved in 60 ears (60%). Scattergrams created according to recently recommended standardized formatting guidelines [23] for the presentation of the direct pre-and postoperative hearing results are presented in Figure 1.

### 3.3. Prediction for Immediate Postoperative Hearing Deterioration or Complete Loss by Influencing Factors

In this model, 77 out of 100 cases were included. For the other 23 cases, no sufficient data was available. The logistic regression model statistically significantly predicted postoperative hearing deterioration χ^2^(9) = 18.176, *p* = 0.033. The model explained 32.2% (Nagelkerke R2) of the variance in postoperative hearing loss and correctly classified 80.5% of cases. Sensitivity was 29.4%, specificity was 95.0%, positive predictive value was 62.5% and the negative predictive value was 82.6%. The area under the ROC curve was 0.829 (95% CI, 0.718 to 0.941), which is an excellent level of discrimination according to Hosmer et al. [24].

Of the nine predictor variables, only two were statistically significant: preoperative BAEP class and resection amount (as shown in Table 4). All other seven variables were not significant. A summary of the Binominal Logistic Regression Analysis is given in Table 4.

### 3.4. Prediction Model for Immediate Postoperative Hearing Deterioration or Complete Loss by Preoperative BAEP and AAO–HNS Class and Discrepant Cases

Patients with worse BAEP scores (BAEP class 3 to 5) carry a high risk of postoperative hearing deterioration or complete loss particularly in combination with a worse AAO–HNS score (Figure 2). The worse the preoperative BAEP class the higher the risk of postoperative hearing deterioration (~50%). Large discrepancies between BAEP and AAO–HNS class are also associated with a high risk of hearing deterioration or loss after surgery.

Twenty-one unusual cases with large discrepancies (≥2 points) between BAEP and AAO–HNS class are listed and highlighted in the Appendix A. Among them, 6 (of 21) ears were functionally deaf in audiometry (AAO–HNS class 4) before surgery but interestingly with available BAEPs. In fact, in these cases, due to the lack of functional hearing, less consideration was given to the unexplained applicable BAEPs and therefore the achieved resection amount was relatively high (in mean 84 ± 18%) as the aim was to remove as much as possible under anatomical preservation the cochlear nerve, to be able to offer the chance of a cochlear implant (or at least an auditory brainstem implant) in these cases.

For the remaining 15 (of 21) ears with preoperative available hearing (AAO–HNS class 1–3) the risk of postoperative hearing loss (AAO–HNS class 4) increases by increasing discrepancy (48% for ≥2 points, 66% for ≥3 points, and 100% for ≥4 points) between BAEP and AAO–HNS class.

### 3.5. Prediction for Immediate Postoperative Hearing Preservation

In patients with preoperative functional hearing (AAO–HNS class 1 and 2), the rate of postoperative hearing preservation was 73% in preoperative BAEP class 1, 67% in class 2, 60% in class 3, 50% in class 4, and zero % in class 5.

### 3.6. Prediction Model for Immediate Postoperative Hearing Deterioration or Complete Loss by Resection Amount Categories

For this model, the data of 95/100 tumors were included in the analysis (Table 5). The remaining 5 tumors had no sufficient data for volumetry-based calculation of the resection amount.

A statistically significant difference in group proportions was seen (*p* < 0.001). Post hoc analysis involved pairwise comparisons using the z-test of two proportions with a Bonferroni correction. The group comparison was statistically significant (*p* < 0.05) between the groups 2 and 5, 3 and 5 as well as between 7 and 5.

## 4. Discussion

Early and frequent monitoring of hearing and tumor volume is indispensable to pursue the goal of median to long-term hearing preservation in patients with NF2. This includes the regular performance of pure-tone and speech audiometry as well as brainstem auditory evoked potentials (BAEP) and thin-sliced cranial MRI for volumetric measurements.

After an initial period of close monitoring (every 3 to 6 months) to assess hearing and tumor dynamics within the first 12 months after initial presentation, monitoring should be continued or extended (every 6 to 12 months) based on patient age, hearing, and tumor growth dynamics as well as tumor size.

Regardless of tumor size, the following patients should be monitored closely: children, adolescents, young adults, patients on bevacizumab therapy, patients with impaired/decreasing hearing, and those with rapidly growing or bilateral medium-sized tumors (T3 according to the Hannover Classification).

Extended follow-up intervals may be considered for small tumors (T1 and T2) in older patients with a slow growth dynamic, associated stable and mildly affected hearing, or in deaf patients.

Except for bilateral large tumors (T4) with critical brainstem compression, the decision for surgical treatment should always be based on impairment of hearing parameters.

Due to their objectivity, the fact that their impairment often precedes audiometry decline, and taking their important intraoperative role into account, BAEPs play a very important and decisive but not exclusive role in the treatment decision.

Their importance for intraoperative monitoring of function in vestibular schwannoma surgery is indisputable and well described [17,18,25].

Generally, there is a positive correlation between BAEP and pure-tone average (or inverse to speech discrimination score) as we described previously [10]. Moreover, the better the hearing before surgery, the higher the chance of preservation postoperatively [10,26]. BAEPs in sporadic VS cases were already described as a predictive factor for postoperative auditory function and the best auditory results are achieved the better the BAEP score (67% and 47% for BAEP class 1 and 2) [18]. Within the context of our very strict hearing preserving criteria, high chances of postoperative functional hearing preservation were achieved the better BAEP scores were (73%, 67%, 60%, and 50% for BAEP class 1, 2, 3, and 4).

Nevertheless, auditory function tests are not to be neglected, particularly for those cases with large (≥2 points) and unusual discrepancies in BAEP and audiometry scoring. For these cases, the risk of hearing loss with increasing discrepancy is between 48 and 100% according to our results. In general, neurophysiological and audiometric results correlate with each other, but in some patients’ large discrepancies were seen either in form of good to excellent BAEP and bad audiometry or vice versa. We observed this rarity in 21% of our investigated ears. However, for the subgroup of special cases with existing and subjectively useful audiometric hearing but pre-and/or intraoperative severely or no more derivable BAEPs, it is worthwhile to continue striving for a more defensive surgical approach in the sense of a partial and then anatomical-guided resection far from the cochlear nerve to maintain the residual chance of audiometric hearing preservation postoperatively.

The mechanism leading to these large discrepancies is unknown and not yet described in the literature. Technical problems or examiner-dependent factors are possible but are unlikely in our setting due to the repeatedly confirmed results, the unchanged equipment, and strictly standard operating procedure driven investigations as well as the inter-examiner reliability.

Certainly, as shown in our current and previous [10] analysis, other factors such as the amount of resection play a role in the prediction of postoperative hearing deterioration [10]. The more resected, the greater the risk of hearing deterioration or complete loss (up to 100% for near-total resection in the current data). Compared to this a devascularization of the tumor by laser coagulation or partial resection amounts carry a significantly lower risk (10–20%). Overall, we were able to achieve higher resection amounts/categories compared to the previous study [10], also due to the more than doubling of preoperative tumor volumes (2 cm^3^ vs. 4.5 cm^3^) in the present cohort, but also poorer postoperative hearing parameters (PTA 21 vs. 44 dB, SDS 81 vs. 69%) and comparable good preoperative audiometric baseline values (PTA 17 vs. 24 dB, SDS 85 vs. 86%). Consequently, too large resection amounts have a negative impact on hearing maintenance in case of NF2-associated VS, so that a less aggressive surgical strategy is recommended with a lower extent of resection amounts.

The mechanisms leading to preoperative hearing deterioration or loss are certainly complex, multifactorial, and not yet fully clarified. Possible mechanisms like mechanical pressure and/or low cochlear perfusion [27,28], hair cell loss due to secretory or regulatory factors [29], as well as cochlear dysfunction because of enhanced elevated levels or perilymphatic proteins [30] are under discussion and needs to be further investigated either by radiological (e.g., special and standardized 3 D fluid-attenuated inversion recovery sequences to depict the perilymph/cochlea [31]), histopathological, or genetic investigation and studies. Furthermore, it is unclear if such factors also influence the immediate postoperative hearing outcome by, e.g., making the cochlear nerve function more or less sensitive to mechanical manipulation.

One limitation of the current study is the likely under detection of cases with postoperative hearing loss. With the presented statistical model, a high specificity but low sensitivity is achieved. This means that postoperative hearing preservation can be predicted with relative certainty based on the data, but a prognosis on postoperative hearing loss remains unreliable. In everyday clinical practice, this residual uncertainty must be discussed with the patients and their relatives.

Patients with discreetly affected but still easily derivable BAEPs, such as class 1 and 2, with simultaneously good PTA and SDS values, have a good chance of direct postoperative hearing preservation in case the surgeon’s intention to treat is hearing preservation. The worse BAEPs before surgery, the higher the risk of postoperative hearing deterioration/loss even in cases with accompanying good PTA/SDS values. As a result, the BAEPs play a very sensitive role in the assessment of postoperative hearing outcomes. As a consequence, we strongly recommend early initiation of surgical therapy for already impaired but well-preserved BAEPs despite normal or slightly impaired audiometry in order to work with good intraoperative BAEPs that allow tailoring the amount of resection according to the intraoperative course of hearing function.

Consequently, each case has to be interpreted and assessed individually. Surgical and neurophysiological expertise plays a very important and decisive role in those special cases with strong discrepancies between neurophysiology and audiometry. Furthermore, in such discrepant cases, unless there is no risk to the patient from a very large tumor (e.g., Hannover Class T4), the results of our study imply, that we will discuss with the patient in advance a surgically rather defensive approach and would recommend tumor debulking with as little manipulation as possible far from the cochlear nerve. In such cases, alternatives such as radiosurgery (for small T1 and T2 tumors), primary bevacizumab treatment (each tumor size), or total resection with a primary hearing restoration such as a cochlea implant or brainstem auditory implant can also be discussed. This is done regularly with all patients that have a bad functional hearing preoperatively where the goal of surgery could otherwise only be to preserve the residual hearing impression.

Compared to our previous study (Gugel et al. 2019 [10]) which investigated the long-term postoperative hearing follow-up in a smaller cohort of young, age-matched NF2 patients, the current study did not confirm the finding of significant reduction of postoperative tumor growth rate or a significant correlation between preoperative tumor volume and postoperative hearing deterioration. Even though not significant, a tendency towards lower postoperative growth rates is seen in the present, non-age-related and older cohort with larger tumor volumes at the time of surgery. Age probably plays an important role in these discrepancies. In young patients, even with small tumors and those with a low growth rate, surgical/therapeutic intervention is carried out at an early stage as soon as hearing deterioration becomes apparent.

A study investigating the short- and long-term follow-up of hearing and tumor growth rate before and after surgery in older NF2 patients is currently undertaken, but there are still not enough valid long-term follow-up data to finish the investigation.

## 5. Conclusions

Both preoperative BAEP and audiometry play a crucial role in decision making for treatment initiation and perioperative hearing monitoring in patients with NF2-associated VS.

Better preoperative values are usually associated with better postoperative auditory outcomes and vice versa. Cases with large discrepancies between BAEP and audiometry are rare but associated with a high risk of postoperative functional hearing loss. For these cases, BAEPs appear to have a higher validity for the auditory outcome of surgery.

## Figures and Tables

**Figure 1 cancers-13-01384-f001:**
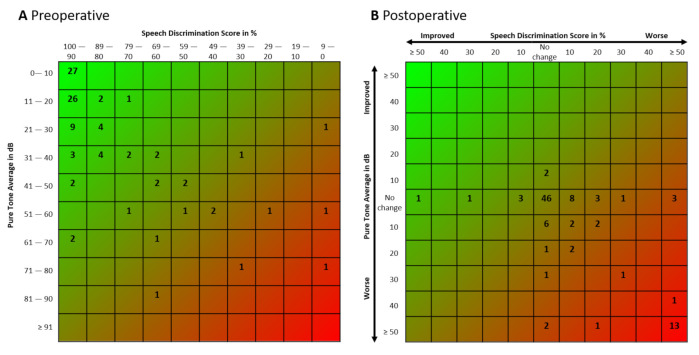
Scattergram of pre-(**A**) and postoperative (**B**) hearing results in NF2-associated vestibular schwannomas (VS).

**Figure 2 cancers-13-01384-f002:**
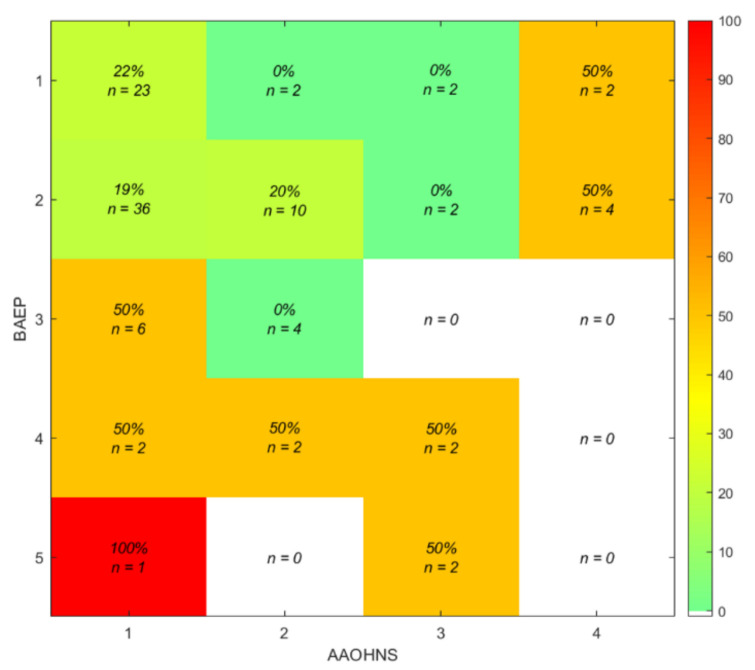
Relative risk prediction of postoperative hearing loss with preoperative BAEP and AAO–HNS class.

**Table 1 cancers-13-01384-t001:** Hannover Classification System of Auditory Evoked Potentials (AEP).

Category	Waves/Latency
1	Waves I, III, and V are present. Latency I–III is normal or slightly increased, (within 2 standard deviations or normal latency, 2.1 ± 0.28 milliseconds (ms).
2	Waves I, III, and V are present. Latency I–III is pathologically increased (>2.66 ms).
3	Waves I and V are present, and Wave III is lost.
4	Wave I is present or Wave V is present.
5	All waves are lost.

**Table 2 cancers-13-01384-t002:** Demographic data of 72 operated neurofibromatosis type 2 (NF2) patients (100 tumors).

Sex (No of female/male)	32/40
Operation side (left/right)	50/50
Family history of NF2 (yes/no)	16/56
Detected mutation types in 36 patients	-
splicing mutations	10
nonsense mutations	6
frameshifting mutations	14
large genome alteration	1
missense mutations	2
large deletions	3
Age at diagnosis in years (mean ± SD, range)	16 ± 9, 1–41
Age at time of surgery in years (mean ± SD, range)	22 ± 9, 8–47
Tumor volume in cm^3^ (mean ± SD, range)	
preoperative	4.51 ± 7.19, 0.11–33.92
postoperative	1.62 ± 2.79, 0–18.05
significance	*p* < 0.001
Growth rate in cm^3^/year (mean ± SD, range)	
preoperative	0.87 ± 2.17, −0.18 to 14.96
postoperative	0.73 ± 2.29, −0.22 to 19.84
significance	*p* = 0.341
Resection amount categories	-
(1) only bony decompression of the IAC	1
(2) decompression of the IAC with laser coagulation (<10%)	10
(3) partial (<10% to <90%)	61
(3a) <10%	12
(3b) ≥10% to <30%	14
(3c) ≥30% to <50%	15
(3d) ≥50% to <70%	15
(3e) ≥70% to <90%	13
(4) subtotal (≥90% to <95%)	3
(5) near total (≥95% to <100%)	7
(6) total (100% including tumor capsule)	6
(7) growth progression	7
Not available	5
PTA in dB (mean ± SD, range)	-
preoperative	24.14 ± 19, 1.25–81.25
postoperative	44.3 ± 42.4, 1.28–130
significance	*p* < 0.001
SDS in % (mean ± SD, range)	-
preoperative	86 ± 23, 0–100
postoperative	69 ± 38, 0–100
significance	*p* < 0.001

No—Number; SD—standard deviation; PTA—pure-tone average; SDS—speech discrimination score; SD—standard deviation. Preoperative values were measured immediately or within 4 weeks before surgery. Postoperative values were measured immediately or up to 3 months after surgery. In all patients, a decompression of the internal auditory canal (IAC) was performed.

**Table 3 cancers-13-01384-t003:** Preoperative and postoperative hearing classifications in 100 operated ears.

G–R Scale [21]	Postoperative Class (No)				
Preoperative Class (No)	I (50)	II (20)	III (12)	IV (0)	V (18)
I (70)	49	8	1	0	12
II (18)	1	11	4	0	2
III (12)	0	1	7	0	4
IV (0)	0	0	0	0	0
V (0)	0	0	0	0	0
**AAO–HNS Classification [22]**	**Postoperative Class (No)**				
**Preoperative Class (No)**	**A (49)**	**B (21)**	**C (6)**	**D (24)**	-
A (68)	48	9	1	10	-
B (18)	0	12	1	5	
C (8)	1	0	4	3	
D (6)	0	0	0	6	
**BAEP Classification System [18]**	**Postoperative Class (No)**				
**Preoperative Class (No)**	**I (18)**	**II (43)**	**III (11)**	**IV (8)**	**V (20)**
I (29)	14	9	1	1	4
II (52)	4	31	6	3	8
III (10)	0	3	4	1	2
IV (6)	0	0	0	3	3
V (3)	0	0	0	0	3

No-Number; G–R—Gardner and Robertson Scale [21]; AAO–HNS Classification—American Association of Otolaryngology–Head and Neck Surgery [22]; BAEP—brainstem auditory evoked potentials Classification System according to Samii and Matthies et al. [18].

**Table 4 cancers-13-01384-t004:** Logistic regression predicting the likelihood of postoperative hearing deterioration or complete loss.

Variables	*B*	S.E.	Wald	*df*	*p*	Odds Ratio	95% CI for B	
							LL	UL
Age at time of diagnosis in years	−0.029	0.045	0.412	1	0.521	0.972	0.891	1.060
Preoperative GR in cm^3^/year	0.018	0.264	0.005	1	0.946	1.081	0.607	1.708
Preoperative volume in cm^3^	0.012	0.077	0.025	1	0.874	1.012	0.870	1.178
Preoperative PTA in dB	0.085	0.048	3.173	1	0.075	1.088	0.992	1.195
Preoperative SDS in %	−0.021	0.034	0.378	1	0.539	0.979	0.915	1.047
Preoperative AAO–HNS class	−1.377	1.292	1.135	1	0.287	0.252	0.020	3.178
Preoperative BAEP class	0.829	0.342	5.871	1	**0.015**	2.292	1.172	4.483
Preoperative G–R class	−1.803	1.194	2.281	1	0.131	0.165	0.016	1.711
Resection amount in %	0.023	0.012	3.901	1	**0.048**	1.023	1.000	1.047

**Note.** **p* < 0.5; B—regression coefficient; S.E.—Standard error; df—difference; CI—confidence interval; LL—lower limit; UL—upper limit; GR—growth rate; PTA—pure-tone average; SDS—speech discrimination score; AAO–HNS Classification—American Association of Otolaryngology–Head and Neck Surgery [22]; BAEP—brainstem auditory evoked potentials Classification System according to Samii and Matthies et al. [18]; G–R—Gardner and Robertson Scale [21]. Significant *p*-values are highlighted **in bold**. Mean values are given in Table 2.

**Table 5 cancers-13-01384-t005:** Prediction model for immediate audiometric postoperative hearing deterioration/loss and preservation vs. resection amount categories.

Resection Amount Category	Immediate Postoperative Hearing Deterioration/Loss	Immediate Postoperative Hearing Preservation
**1**	*n* = 0 (0%)	*n* = 1 (100%)
**2**	*n* = 1 (10%)	*n* = 9 (90%)
**3**	*n* = 12 (20%)	*n* = 49 (80%)
**4**	*n* = 2 (67%)	*n* = 1 (33%)
**5**	*n* = 7 (100%)	*n* = 0 (0%)
**6**	*n* = 3 (50%)	*n* = 3 (50%)
**7**	*n* = 0 (0%)	*n* = 7 (100%)
**Total**	*n* = 25 (26%)	*n* = 70 (74%)

*n* = numbers of tumors. Numbers of tumors and resection amount category description are given in Table 2.

## Data Availability

All data is included in this manuscript and the Appendix A.

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
