# Peer review of "Risk Stratification for Immediate Postoperative Hearing Loss by Preoperative BAER (Brainstem Auditory Evoked Response) and Audiometry in NF2-Associated Vestibular Schwannomas"

_cancers, 2021, doi:10.3390/cancers13061384_

Round 1
Reviewer 1 Report
I enjoyed reading this manuscript on a very important topic to the NF2 community, and one that is difficult to manage, so the input from this experience group is valuable. There are a number of concerns which could be addressed to help clarify the content considerably.
The newer AAO-HNS criteria for reporting hearing change in clinical studies have replaced the old criteria referenced in this study and provide a much more complete idea of what hearing changes have occurred. See Gurgel RD, et al, Otolarygol HNSurg 2012 Nov;147(5):803-7. Doi 10.1177/0194599812458401. The old AAOHNS criteria and the Gardner – Robinson scale may still be referenced, but are so much less clear.
- Experience of surgeon was not noted, but may be an important factor contributing to variability in outcome to consider. How many different surgeons participated in this study?
- Was the hearing used to report post op values in Table 2 the earliest, the latest, or the best post op value. How did these compare? A full file of this data along with volumetric data and intraoperative data in supplemental form would add great value to the study.
- Knowledge of where all patients started individually within each class could be enlightening. A patient with poor hearing, for example might have poor AAOHNS or G-R class preop and have no change in their PTA or SDS score. What criteria is used to sum up the change from surgery?
- The amount of tumor resection needs to be clarified. At one point it appears as a percent resected, but in other places it is only noted that 61 of the patients were in the Partial Resection group. Again, a table of individual values would be helpful.
- Line 68 “is less aggressive” than what?
- Line 69 Define “radical” surgery. The authors need to clarify the role of continued tumor growth to length of hearing preservation in incomplete tumor removal. In a prior paper the rate of hearing loss post partial resection in patients with initial hearing preservation appeared very similar to the rate of hearing loss in patient who had no tumor resected. Is this the case in this cohort? How many patients from the prior publications in 2019 were included in this study?
- Line 86: The NF2 criteria of Baser et al. are not the same as NIH criteria. Which was used?
- Line 85: 72 NF2 patients with 100 tumors are included in this study. How many were over age the of 25? Were the 39 tumors operated on in the authors Cancers 2019 article included in this study? Was the increase number of tumors in a one year period (2004-2019) to 100 tumors in this study from 39 tumors (2004-2018) in the previous study which had one less year of patient accrual the result of including all NF2 patients rather than just those 25 and younger?
- Immediate maintenance of functional hearing was reported in the earlier group of less than 25 year of age at 82% but on average only 50% of tumor volume was resected. This cohort appears to have had greater percent tumor resection by volume decrease reported. Why was there such a difference in the amount of tumor resected in this cohort? Was there a change in philosophy or indication for surgery over the year with mean post op 1 cm3 vs preop 2 cm3 tumor volumes in the prior study to 1.62 cm3 post op mean to 4.51 cm3 pre op mean in the current study. Larger tumors seems to be evident in this cohort, is that correct?
- Table 1. What was the reported pre and post op tumor growth rate in partially resected tumors compared to non-operated contralateral VS tumors within patients. What was the growth rate when compared to patients who did not undergo surgery. Is it possible to match for age, tumor size and hearing status in the unoperated group from prior reports.
- Line 149: What is the variance in postoperative hearing loss determined by (PTA, SDS, G-R, AAOHNS) to determine the PPV for hearing deterioration? Could this be clarified so a clinician could use to counsel a patient from this data?
- Line 133. Just a minor observation that “nearly 60% were partially resected” but 61 of 100 were partially resected in table 1.
- What difference was there in hearing preservation and growth rate in the tumors that were only decompressed or devascularized vs those that were partial or subtotally resected.
- How many patients went on bevacizumab following incomplete resection?
- Was decision made for partial resection only on BAER intraoperatively or was the hearing status in the opposite ear a consideration? Were patients with poor speech discrimination more likely to have complete removal regardless ob BAER status?
- Line 86 stipulates 72 NF2 patients and 100 tumors with complete hearing diagnostic before and after surgery are included in this retrospective analysis – but then (line 137) states that only 73 ears were evaluated for post-operative hearing and 77 ears on line 147. What happened to the other 27 (or 23) ears? These statements are confusing.
- Preservation of the preoperative hearing in 73% is for both PTA and SDS? What if PTA decreased and SDS held constant? Was that scored as decrease? Stating what the BAEP classification is, would be useful. It is reference to an 1997 paper by Matthies et. al. Is that still the standard? If so, clarifying what the scale is and how it applies here would be useful to the reader. Is this the Hanover scale?
- Indications for surgery in prior reference a) large tumor T4 Hannover Classification on both sides, b) continuing tumor growth and deterioration of auditory evoked potential or impairment of PTA or Speech discrimination score (SDS) during observation. Which criteria applied in this study. Both growth and hearing deteriorization of one or more? (ABR, PTA, SDS)
- How were volumetric measurements made? Reference is given to iPlan Net software (Brainlab) vs manual segmentation. Is this still the method used or both? Partial resection ranged from <10% to 90% a-e categories in 2019 J Neurosurg Pediat. How did these patients rate on the resection scale? Was this determined by surgeon or volumetrics? Do Subtotal (90-95%), and Near Total 95 to <100% still hold in this analysis. Briefly stating the current volumetric analysis method would be helpful and clarifying the %volumes.
- Did any patients in this study already have a surgery prior to study entry? e. had continued tumor growth prompted a second surgery? How many of the partially resected tumors were reoperated?
- Were any tumors operated at outside institutions?
- How many different surgeons and years of experience?
- Line 108: On data evaluation, it is unclear where a patient would fall with stable PTA pre- and post op and a decrease >15% SDS compared to pre-op values. Should it state decrease of >15db in pure-tone average (PTA) or >15% in speech discrimination?
- Line 114 are AAO-HNS, and G-R classifications not dependent upon PTA and SDS? As continuous variables, the latter are more rich in data than the clumped former. Change in PTA and SDS for each patient is much more meaningful than change in category (AAO-HNS old criteria or G-R) as a the boundaries are artificial and wide as noted above.
- Lines 117-121. I am unfamiliar with the Box-Tidwell procedure or with the decision to retain or reject values based on the standardized residuals. Perhaps review by a statistician would be helpful here.
- It would be enlightening to comment on why is there no difference in growth rate pre and post operatively in this study compared to previous study where operated tumors grew more slowly than they had preoperatively.
- How many patients were treated with post-operative bevacizumab or other chemotherapeutics?
- Line 155. If only two variables were found to be statistically significant (BAEP class and resection amount), it might be enlightening to know how much was resected in the “partial resection” category as the range was very wide in the reference given as noted above. Partial resection ranged from <10% to 90%. Was the exact change in volume used for the regression or only the category of resection?
- How were the BAEP’s performed? Were adjustments made to classification of BAEP category based upon the degree of loss of PTA?
- In patients 2.2 and 7.2, were total resections carried out because preoperative SDS was poor (30% and 20% respectively) without regard for the BAEP class? Which holds more weight in the surgeons decision making process?
- Were there any cases where intraoperative BAER response was lost, but he patients hearing was retained?
- Was there any correlation with intraoperative BAER and post operative hearing or was only preop BAER considered?
- Table 5: Difficult to read as several values don’t line up well with descriptors.
- Line 195-198: The meaning of this sentence in not clear. Also 199 – 200 is unclear. At what age are adults considered for less frequent BAER monitoring? What size constitutes small tumors, etc.
- Line 200: Why are “treatment decision…mainly based on hearing values” if BAER is one of the determinants significant in hearing preservation, but PTA and SDS were not?
- Line 200-201: Needs to be clarified.
The length of follow up needs to be discussed. Is there benefit to early pre-hearing loss excision. Early hearing preservation is a worthy goal if it persists, but if it does not significantly lengthen the time of useful hearing, and the tumor has to be re-excised later, does it make sense to wait until the hearing loss has occurred before resecting the tumor?
An important point of discussion not mentioned, but relevant is the potential use of a cochlear implant in the case of preservation of the cochlear nerve, but loss of function. The overall response in limited retrospective case series indicate much better speech understanding with CI’s than ABI’s. Early intervention may be relevant in this situation.
Reviewer 2 Report
This is an interesting study investigating the risk of postoperative hearing loss after surgery for VS in NF2 patients by investigating BAEP, audiometry and clinical data.
- Introduction (line 63): Stereotactic radiosurgery (SRS) is also an option for the treatment of NF2-associated VS. A recent systematic review of 974 NF2-associated VSs showed that SRS demonstrated higher rates of local control and significantly lower facial nerve complications (PMID: 28882713).
- The logistic regression model has a sensitivity of 29.4 % which indicates an under-detection of postoperative hearing loss. However, the AUC is pretty good. The authors should discuss the limitations of their statistical model.
- The manuscript lacks a discussion with regard to some potential pathophysiological causes (e.g. Cytokines, heat shock protein 70) for postoperative hearing loss next to the known predictors such as extent of resection and preoperative hearing function. The literature discusses the importance of increased concentrations of perilymphatic protein concentrations and subsequent increased cochlear signal intensity on FLAIR magnetic resonance images (PMID: 24742808). Have the authors analyzed this imaging characteristic in their VS series?
- In the discussion (line 226) the authors suggest that preoperative tumor volumes play a significant role in the prediction of postoperative hearing deterioration. However, this finding of their previous study investigating a smaller cohort could not be confirmed in the present study. This should be corrected and explained/discussed.
- It would be interesting to know some histopathological data. Can we authors describe the amount of macrophage infiltration in the included VSs. For instance, the literature debates a trend toward poorer hearing function in patients with higher amounts of tumor-associated macrophages (PMID: 31430634).
- The authors present a lot of data and describe their methods extensively. Yet, at the end of the article, it remains somewhat unclear what the message of this work actually is. What exactly are the results supposed to serve in clinical practice and what added value do they have for future studies or clinical therapy. Here, an extensive revision of the manuscript is recommended.
- Minor issue: Line 147 of the section results: I think the authors mean 23 patients or were 110 patients analyzed for this objective?
Round 2
Reviewer 1 Report
Thank you for your extensive revisions. Addition of the new figures and tables are helpful.
I believe point 18 might be worthy of inclusion in the manuscript because when the BAER is lost intraoperatively some surgeons may be more aggressive at tumor removal but your recommendation to be more conservative is of value. Point 36 might also be included for the same reasoning.
Point 39. I apologize for misunderstanding the "hearing values" included the BAER testing. The later is more of a physiologic measure than subjective hearing. Thank you for clarifying this point.
